# What Can One Minute of the Day Tell about Physical Activity?

**DOI:** 10.3390/ijerph20196852

**Published:** 2023-09-28

**Authors:** Henri Vähä-Ypyä, Pauliina Husu, Harri Sievänen, Tommi Vasankari

**Affiliations:** 1UKK-Institute, 33500 Tampere, Finland; pauliina.husu@ukkinstituutti.fi (P.H.); harri.sievanen@ukkinstituutti.fi (H.S.); tommi.vasankari@ukkinstituutti.fi (T.V.); 2Faculty of Medicine and Health Technology, Tampere University, 33014 Tampere, Finland

**Keywords:** physical activity intensity, cardiorespiratory fitness, 24/7 measurement

## Abstract

High cardiorespiratory fitness (CRF) allows individuals to perform daily activities and operate at a higher intensity level. This study investigates the connection between the CRF and peak intensity of physical activity (PA) in absolute and relative terms. A total of 3587 participants (1447 men, 51.9 ± 13.0 years; 2140 women, 50.0 ± 13.0 years) provided substantial accelerometer wear time, and their CRF was estimated via the 6 min walking test. Participants were divided into CRF thirds by age group and sex. Daily one-minute peak intensities were captured in both absolute terms and relative to individual CRF levels. In absolute terms, the highest CRF third had the highest intensity value for men (6.4 ± 1.7 MET; 5.9 ± 1.4 MET; 5.3 ± 1.0 MET) and for women (6.4 ± 1.6 MET; 5.9 ± 1.3 MET; 5.4 ± 1.1 MET). In relative terms, the highest CRF third utilized the least aerobic capacity for men (49 ± 14%; 51 ± 13%; 56 ± 14%) and for women (52 ± 13%; 54 ± 12%; 62 ± 15%). One minute of daily activity offers valuable insights into an individual’s CRF and the effort demanded during PA. Fitter individuals can sustain higher PA intensity levels in absolute terms, whereas individuals with lower CRF utilize a greater fraction of their aerobic capacity. Consequently, heightened CRF not only allows for enhanced intensity levels but also safeguards against strenuous PA during daily routines.

## 1. Introduction

Cardiorespiratory fitness (CRF) and physical activity (PA) are important for overall health and well-being [1]. CRF is associated with better overall health and a reduced risk of developing chronic diseases such as cardiovascular disease, diabetes, certain types of cancer, and obesity [2,3]. PA is essential for maintaining a healthy lifestyle and has numerous benefits, such as improving cardiovascular health, maintaining a healthy weight, reducing the risk of chronic diseases, and improving the overall quality of life [4,5]. PA can consist of aerobic, muscle-strengthening activities, balance enhancing, and physical function-enhancing activities [5]. The principal modifiable determinant of CRF is a habitual PA level besides age, sex, health status, and genetics [1]. Therefore, affecting PA can be considered a means to influence CRF as well.

CRF reflects the integrated ability of the body to transport oxygen from the atmosphere to the mitochondria in cells of skeletal muscles to perform physical work [6]. Higher CRF levels are associated with better cardiovascular function, including increased cardiac output, improved blood flow, and better oxygen delivery to working muscles [6]. This allows the body to efficiently meet the increased demand for oxygen during strenuous activity [7]. CRF is directly related to the integrated function of numerous systems, and it is thus considered a reflection of health [8]. The main parameter that reflects CRF is maximal oxygen uptake (VO_2_max) [6].

The intensity of aerobic PA can be expressed in either absolute or relative terms [9]. Absolute intensity is the amount of energy expended during the given activity without considering a person’s CRF or aerobic capacity. Moderate-intensity physical activities (MPA) have a metabolic equivalent (MET; 1 MET = 3.5 mL(O_2_)/kg/min) value of 3–5.9 METs, vigorous-intensity physical activities (VPA) have a MET-value of 6–8.7 METs, and very vigorous-intensity physical activities (vVPA) have a MET-value of 8.8 METs or greater [10]. In contrast to absolute intensity, relative intensity denotes the level of effort relative to an individual maximum aerobic capacity [9]. The relative intensity can be estimated using the percentage of oxygen uptake reserve (VO_2_R), VO_2_max, heart rate reserve or maximum heart rate [10]. In terms of VO_2_R, MPA has a VO_2_R value of 40–59%, VPA a VO_2_R value of 60–89% and vVPA a VO_2_R value of 90% or greater [10].

Typically, the device-based assessment of PA is based on population-specific cut points, which categorize the measured absolute intensity into light PA, MPA, and VPA [9]. The measurements based on the absolute intensity can provide comparable results across populations, but they ignore the individual’s own exercise capacity or activity type preferences [11]. Individuals with high CRF may naturally engage in more vigorous physical activity and maintain higher absolute intensities during ambulatory activities and can engage in activities requiring more exertion and higher intensity. In PA measurements, individuals with higher CRF have higher daily moderate-to-vigorous (MVPA) and VPA time accumulation in absolute terms than individuals with lower CRF [12]. They also have higher adherence to PA guidelines in absolute terms [13]. In relative terms, when PA is classified according to individual fitness levels, the results become complex. Individuals with lower CRF accumulate more MVPA and VPA time, especially from short lengths of activity [12], and have higher adherence to PA guidelines [13] than individuals with higher CRF. In relative terms, the association between the VO_2_max and accumulated MVPA and VPA times is positive only for the long, 60 min MVPA or 30 min VPA bouts [12]. Therefore, individuals with high CRF have a greater capacity for sustained aerobic activity, and individuals with low CRF have to utilize most of their aerobic capacity to cope with their daily routines.

Although a high CRF is an important indicator of overall health and is associated with numerous health benefits, CRF remains a vastly undervalued measure in both clinical settings and in the public health arena [2]. There are no general recommendations for CRF [6], which is in contrast to PA [5,14]. In adults, a CRF level of less than 5 METs is associated with a high risk for mortality, but a sufficient CRF level may vary depending on age, sex, and individual factors [8]. Regarding differently aged men and women, the respective minimum CRF levels associated with a significantly lower risk of cardiovascular events are 9 and 7 METs for 40 years old, 8 and 6 METs for 50 years old, and 7 and 5 METs for 60 years old [15]. Anyone can experience health benefits from improving their CRF, but the benefits are most evident at the low end of the CRF continuum [8]. Relatively small changes in CRF can have noticeable health effects for the least fit individuals [1,16].

Recently, it has been suggested that device-based measurements of physical behavior/PA and CRF should be synchronized [17]. Instead of the commonly reported accumulated daily aerobic PA time, a better option might be to measure the peak intensity of the daily aerobic PA [18]. Combining measurements of CRF and free-living peak aerobic PA intensity allows for a holistic understanding of both the individual’s aerobic capacity and their PA relative to their capacity during their daily life [9]. For example, measured peak intensity has previously been found to differ between the cardiovascular disease groups when the participants were categorized into three groups according to the Framingham risk score [19]. The purpose of the present study is to further investigate the association between the measured CRF, and the highest intensity of aerobic PA reached in a population sample of working-age adults. Additionally, this study aims to define the CRF threshold above which individuals can effectively carry out their daily routines.

## 2. Materials and Methods

This study is based on a subsample of the population-based FinFit2017 and FinFit2021 studies [20,21]. The FinFit2017 and FinFit2021 are cross-sectional population studies based on a stratified random sample of 20–69-year-old Finnish men and women from seven city-centered regions. The subsample comprised 3587 participants (1447 men and 2140 women), aged 20–69 years, who completed a 6 min walk test (6MWT) and wore an accelerometer 24 h per day for at least four days during a one-week data collection period. The co-ordinating ethics committee of The Regional Ethics Committee of the Expert Responsibility Area of Tampere University Hospital gave ethical approval for the study (R17030 and R21050).

During the health examination of the study, participants’ height and weight were measured, and they performed a 6MWT [22]. In short, the participants were asked to walk back and forth along a 15 m walking track as fast as possible for six minutes. Their heart rate was recorded with a heart rate monitor (Polar M61, Polar Electro, Kempele, Finland). For men, the VO_2_max was predicted from the walking distance in six minutes, age, body mass index (BMI), heart rate at the end of the test, and height [22]. For women, the heart rate was not a statistically significant predictor, and the prediction was based on the walking distance in six minutes, body mass, and age [22]. These predictors explained 82% (standard error of estimate (SEE) = 3.6 mL/kg/min) of the variation in the measured VO_2_max in men and 79% (SEE = 3.5 mL kg/min) in women [22].

At the health examination, the participants also received a triaxial accelerometer (UKK RM42, UKK Terveyspalvelut Oy, Tampere, Finland) to be used for seven consecutive days all the time except during showers and other water activities. The device was attached to a flexible belt with an instruction to wear the belt so that the accelerometer was on the right hip during waking hours and on the nondominant wrist during time in bed. The acceleration signal was collected at a 100 Hz sampling frequency, ±16 g acceleration range (g is Earth’s gravitational constant), and 0.004 g resolution. After the one-week measurement, the accelerometers were returned, and the raw data were stored on a hard disk for further analysis.

The raw accelerometer data were analyzed in 6 s epochs according to our standard procedure [23]. For each epoch, the mean amplitude deviation values of the resultant acceleration signal and of the acceleration signal in each orthogonal direction were calculated [23]. The epoch-wise acceleration values were converted to METs [12]. The accuracy of the MET estimation is about 1.2 MET for bipedal locomotion over a wide range of speeds [23].

The epoch-wise MET values were further smoothed by a 1 min exponential moving average filter, and the highest measured value during the calendar day was recorded as the daily peak MET value. The daily peak MET value was recorded in both absolute values and relative to an individual’s VO_2_R. The VO_2_R denotes the reserve between the resting and maximum oxygen uptake level [10], and it was calculated with the equation VO_2_R = (peak MET − 1)/(VO_2_max/3.5 − 1), where 3.5 denotes the oxygen consumption of the metabolic equivalent. The average and standard deviation (SD) of the daily peak MET values were computed for each participant’s measurement week. These values were then utilized to calculate the individual coefficient of variation (CV) for the measurement week. Additionally, the measurement days were divided into 24 segments, i.e., 1 h segments, and the peak MET value was recorded for each segment. The number of segments where the peak MET value represented at least moderate PA or 40% of VO_2_R, vigorous PA or 60% of VO_2_R, or very vigorous PA or 90% of VO_2_R was calculated in absolute and relative terms.

To examine whether CRF modulates the daily peak intensity value in both absolute and relative terms, the participants were categorized into tertiles and deciles according to their VO_2_max results. The division into tertiles was carried out separately for men and women in five age groups: 20–29 years, 30–39 years, 40–49 years, 50–59 years, and 60–70 years (nine participants had already reached the age of 70 years by the time they took part in the tests). The division into deciles was also carried out separately for men and women, and the participants were weighted by the sample size in the previous age groups.

Partial Spearman correlations controlled for age, age^2^, and sex were calculated to quantify the degree of association between peak MET and peak VO_2_R values and VO_2_max. The differences between the VO_2_max values, BMI values, peak MET values, and peak VO_2_R values in the CRF tertiles were tested separately for men and women with the independent-samples Kruskal–Wallis test with a posthoc Dunn’s test adjusted by the Bonferroni correction for multiple tests. The differences between the number of active hours, peak MET values, peak VO_2_R values, and CV values in CRF deciles were tested separately for men and women with an independent-samples Kruskal–Wallis test with a posthoc Dunn test adjusted by the Bonferroni correction for multiple tests. The significance level was set at *p* < 0.05. All analyses were conducted by IBM SPSS version 29.0.

## 3. Results

Broken down by sex and age group, Table 1 shows the VO_2_max, BMI, mean daily peak MET value, and VO_2_R in the CRF tertiles. The highest BMI values were observed in the low CRF group and the lowest in the high CRF group. The differences between the groups were significant except for the younger men. In absolute terms, the highest peak MET was observed in the high CRF group, and the lowest values in the low CRF group. The mean peak MET values from the highest to the lowest CRF group were 6.4 (SD 1.7), 5.9 (1.4), and 5.3 (1.0) for men, and 6.4 (1.6), 5.9 (1.3), and 5.4 (1.1) for women. The men’s and women’s peak MET values were at the same level. The difference between the groups was significant for the older age groups. In relative terms, low CRF groups reached the highest proportion of the oxygen uptake reserve, and the differences between the CRF groups were significant for the older age groups. The mean peak VO_2_R values from the highest to the lowest CRF group were 48.8% (SD 13.8%), 51.2% (12.6%), and 56.4% (14.0%) for men, and 51.7% (13.5%), 54.0% (12.3%), and 62.4% (16.4%) for women. Based on the peak VO_2_R values, the women had to utilize more of their aerobic capacity than men. The older participants in the high CRF group had similar VO_2_max and daily peak MET values than the younger participants in the low CRF group. In absolute terms, the men and women were moving at the same intensity, but in relative terms, the women utilized more of their aerobic capacity. The VO_2_max correlated significantly (*p* < 0.001) with the peak MET value (r = 0.379) and the peak VO_2_R value (r = −0.405).

Figure 1 shows the mean BMI and age in the CRF deciles according to sex. The BMI and age decrease with increasing VO_2_max for both men and women. The significance of differences in BMI and age values between the CRF groups can be found in Appendix A.

The daily peak MET and VO_2_R values are shown in Figure 2. In absolute terms, the peak MET value increases with increasing CRF. In relative terms, the peak VO_2_R value decreases with the increasing CRF. The significance of differences in peak MET and VO_2_R values between the CRF groups can be found in Appendix A.

Figure 3 displays the mean of the individual coefficient of variation (CV) of the daily peak MET values. The individual CV increases with increasing CRF (Appendix A) despite the greater mean value (i.e., the denominator of CV). Consequently, as CRF decreases, the days tend to become more monotonous in terms of the intensity of PA.

Figure 4 shows the number of one-hour segments containing at least one bout, where the peak MET was over 3 MET or 40% of VO_2_R. In absolute terms, the number of segments increases with increasing VO_2_max. In relative terms, the trend is inverted, and the number of segments increases with decreasing VO_2_max. For both men and women, the lowest CRF decile exhibits a significantly higher number of active hours compared to the other deciles (Appendix A). Men and women showed a similar trend in both absolute and relative terms.

Figure 5 shows the number of one-hour segments containing at least one bout, where the peak MET was over 6 MET or 60% of VO_2_R. In absolute terms, the number of segments increases with increasing VO_2_max. In relative terms, the trend of the mean values is U-shaped. The lowest CRF deciles have the highest number of segments with peak MET over 60% of VO_2_R (Appendix A).

Figure 6 shows the number of one-hour segments containing at least one bout, where the peak MET was over 8.8 MET or 90% of VO_2_R. In absolute terms, the number of segments rises as VO_2_max increases, with the mean values displaying a distinct change in direction around the VO_2_max level of 30 mL/kg/min. In relative terms, although the mean values appear to follow a U-shaped trend, there are no significant differences in values between the lower and higher deciles when compared to the middle deciles (Appendix A).

## 4. Discussion

The findings of this study indicate that participants with high CRF are willing to exercise, while participants with low CRF have to utilize a greater fraction of their aerobic capacity during daily activities. The high CRF group exhibited the highest absolute intensity values, whereas the low CRF group achieved the highest relative intensity values during their daily routines. The participants with high CRF also had a higher number of active hours containing moderate, vigorous, or very vigorous PA bouts when the activity was classified in absolute terms. On the other hand, in relative terms, the participants with low CRF had the highest number of hours containing moderate PA bouts. For vigorous and very vigorous activity, a U-shaped behavior was evident in relative terms. Alongside the positive outcomes, an acute PA bout also causes a stress response in the body depending on the duration and intensity of the bout [7,24]. The U-shaped outcome in relative terms (Figure 5 and Figure 6) can indicate that participants with low CRF can have vigorous PA bouts and consequent stress responses throughout the day. In contrast, high CRF can protect individuals from strenuous PA during daily activities [6].

Individuals with low cardiorespiratory fitness (CRF) can be classified into two groups–those leading a sedentary lifestyles and those who engage in physical activity within their capacity [11]. Those engaged in sedentary lifestyle have substantial potential to enhance their CRF by increasing their physical activity levels [16]. On the other hand, individuals in the latter group might exhibit high activity levels in relative terms, but their low fitness persists apparently due to insufficient time for recovery [13]. It is worth noting that leisure-time physical activity has been found to have a beneficial effect on CRF, while occupational physical activity may have an unfavorable impact [25,26]. Leisure-time physical activity often includes dynamic movements at conditioning intensity levels sufficient to improve CRF over short time periods with enough recovery time. Occupational physical activity, in turn, constitutes, in general, too low of intensity to have any substantial impact on the CRF level without sufficient recovery time [27,28]. Adding extra leisure-time exercise may not be feasible for individuals who are exhausted after a working day [28].

In this study, individuals with a VO_2_max of less than 30 mL/kg/min demonstrated the highest number of hours containing vigorous physical activity bouts in relative terms. As a result, they have to utilize a high fraction of their aerobic capacity throughout the day. Given that CRF tends to decline with aging, individuals with a VO_2_max of slightly over 30 mL/kg/min should be encouraged to maintain or further increase their CRF through regular exercise [11]. While the absolute results of this study indicate that both men and women exhibit similar peak MET values, the relative findings suggest that women have to utilize a higher proportion of their aerobic capacity compared to men. As a result, further research is warranted to explore those factors influencing the determination of intensity in daily routines, including sex disparities and variations based on individual fitness levels.

The observed U-shaped pattern in relative terms (Figure 5 and Figure 6) also suggests that individuals with high CRF engage in exercise voluntarily and intentionally. However, it is essential to note that the analysis algorithms used in the present study were validated specifically for bipedal activities [23]. As a result, the intensity of activities such as cycling and cross-country skiing may have been underestimated and muscle-strengthening or balance-enhancing activities are not captured properly. Moreover, participants were asked to remove the device before water-based activities such as swimming to avoid discomfort due to a wet elastic belt, although the device itself was water-resistant. Without these restrictions, the number of activity bouts in relative terms would likely have been higher for participants with high CRF, making the U-shaped pattern even more evident. Furthermore, it is important to acknowledge that the precision of our VO_2_max and peak MET estimations could be considered a limitation of this study. Achieving greater precision in VO_2_max measurements would require a test to be carried out until exhaustion and involving breath-by-breath measurements of the gas concentrations of inspired and expired air [22]. Enhancing the precision of the peak MET estimation could involve individual calibration [27]. Nevertheless, considering the large-scale sample size in the present study, the methods employed can be viewed as both feasible and cost-effective.

While blue-collar workers are physically active for several hours each working day, they tend to have poorer cardiorespiratory fitness (CRF) and overall health compared to white-collar workers [29]. This raises the question of whether all types of PA are equally health-enhancing [28]. The recent WHO guidelines for physical activity and sedentary behavior acknowledge the need for further research on the impact of different domains of PA on health [5]. It is essential to consider that adults with high levels of occupational physical activity might benefit from more detailed recommendations for leisure-time physical activity [28]. While they may already have high levels of mandatory physical activity, encouraging them to increase their activity levels further could potentially result in chronic strain, injuries, or fatigue. [30]. They could benefit from the optimal amount of PA, sedentary behavior, and recovery to improve their CRF level [29]. Occupation-specific recommendations for CRF may also be necessary.

Beyond considering the domain of PA, individual CRF deserves greater attention [31]. A single exercise bout has both positive effects and negative, fatiguing impacts on the body’s homeostasis [6]. A higher CRF can mitigate the negative effects during daily routines and occupational tasks. High levels of CRF are associated with a lower risk of cardiovascular diseases, metabolic disorders, and mortality, while low CRF is linked to an increased risk of adverse health outcomes [2]. The impact of relatively small CRF changes on mortality risk has considerable clinical and public health significance [32,33]. Further evidence is needed to establish cut points or thresholds that determine low, moderate, and high CRF across various age, sex, and race groups [8]. Based on our current study, a VO_2_max of approximately 30 mL/kg/min appears to protect from strenuous PA during daily routines. Low-fitness individuals have to work at a higher percentage of their aerobic capacity to perform the same activities as their more fit counterparts, which can put additional strain on their cardiovascular system [33]. Moreover, they might find it challenging to sustain vigorous efforts during physical activity and become fatigued more quickly [34]. Nevertheless, it is advisable to take proactive actions to mitigate the decline in CRF well before reaching this critical threshold.

## 5. Conclusions

One minute of the day can provide valuable information about the individual’s CRF and the level of effort required during habitual PA. The highest measured absolute one-minute peak MET level indicates higher CRF, as more fit individuals are able to sustain higher intensity levels of PA. When the individual’s CRF is taken into account, individuals with lower CRF have to utilize more of their aerobic capacity in the same activities than those with higher fitness levels. Thus, high CRF permits one to operate at a higher intensity level and protects from strenuous PA during daily routines. Based on the present study, a VO_2_max of at least 30 mL/kg/min can protect from strenuous PA during daily routines.

## Figures and Tables

**Figure 1 ijerph-20-06852-f001:**
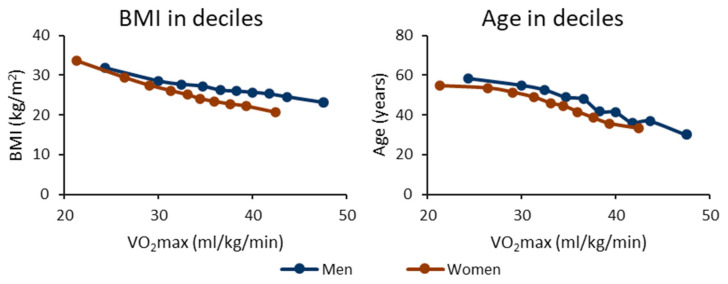
Body mass index (BMI) and age in cardiorespiratory fitness deciles. The *y*-axis shows the mean BMI or age value, and the *x*-axis shows the maximal oxygen uptake (VO_2_max) of the decile. The blue line shows the men’s value, and the red line shows the women’s value. The vertical whiskers show a 95% confidence interval.

**Figure 2 ijerph-20-06852-f002:**
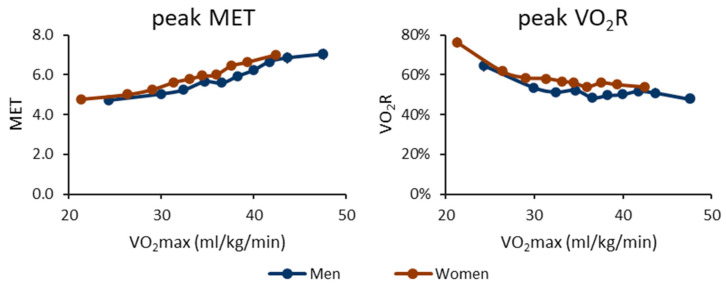
Daily peak intensity for metabolic equivalents (MET) and oxygen uptake reserve (VO_2_R) for cardiorespiratory fitness deciles. The *y*-axis shows the mean peak MET or peak VO_2_R value, and the *x*-axis shows the maximal oxygen uptake (VO_2_max) of the decile. The blue line shows the men’s value, and the red line shows the women’s value. The vertical whiskers show a 95% confidence interval.

**Figure 3 ijerph-20-06852-f003:**
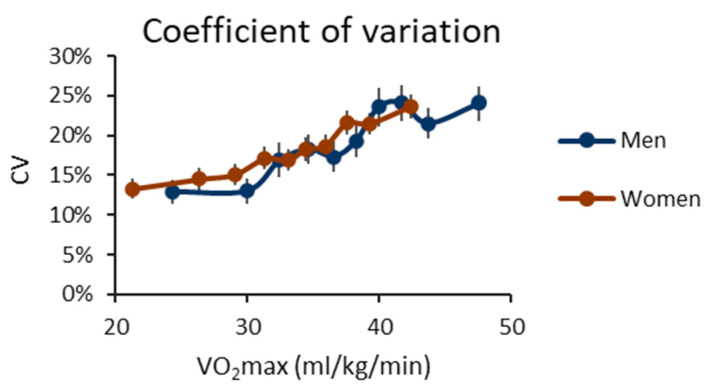
The mean induvial coefficient of variation (CV) of the daily peak intensity for cardiorespiratory fitness deciles. The *y*-axis shows the CV value, and the *x*-axis shows the maximal oxygen uptake (VO_2_max) of the decile. The blue line shows the men’s value, and the red line shows the women’s value. The vertical whiskers show a 95% confidence interval.

**Figure 4 ijerph-20-06852-f004:**
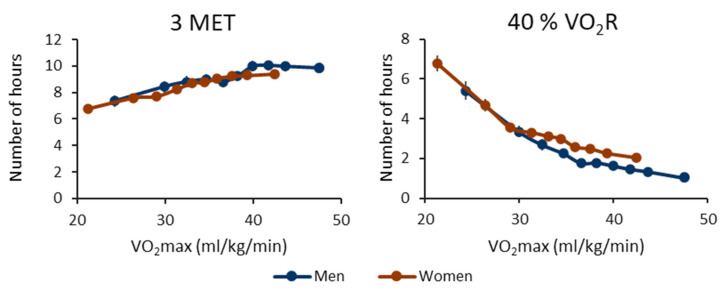
The number of hours containing at least one moderate physical activity bout for the cardiorespiratory fitness deciles. The left panel shows the results for metabolic equivalents (MET), and the right panel shows the oxygen uptake reserve (VO_2_R) value. The threshold for a moderate bout is 3 MET or 40% of VO_2_R. The *y*-axis shows the number of active hours, and the *x*-axis shows the maximal oxygen uptake (VO_2_max) of the decile. The blue line shows the men’s value, and the red line shows the women’s value. The vertical whiskers show a 95% confidence interval.

**Figure 5 ijerph-20-06852-f005:**
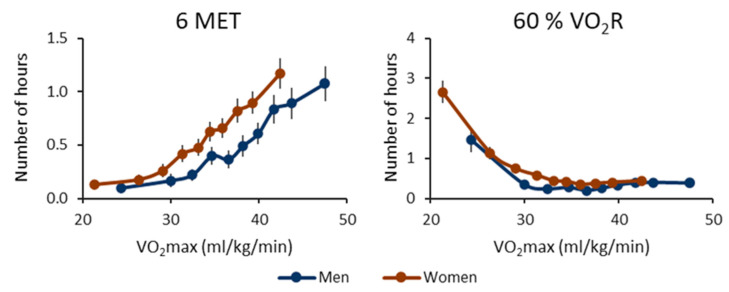
The number of hours containing at least one vigorous physical activity bout for the cardiorespiratory fitness deciles. The left panel shows the results for metabolic equivalents (MET), and the right panel shows the oxygen uptake reserve (VO_2_R) value. The threshold for a moderate bout is 6 MET or 60% of VO_2_R. The *y*-axis shows the number of active hours, and the *x*-axis shows the maximal oxygen uptake (VO_2_max) of the decile. The blue line shows the men’s value, and the red line shows the women’s value. The vertical whiskers show a 95% confidence interval.

**Figure 6 ijerph-20-06852-f006:**
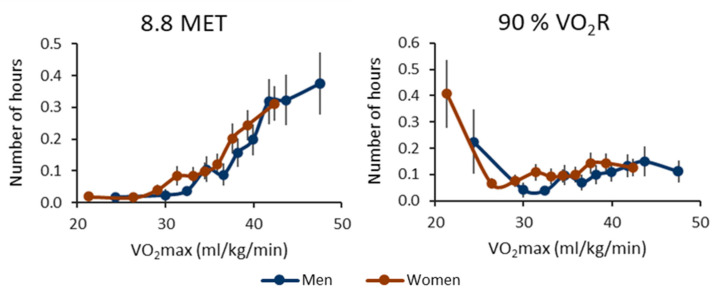
The number of hours containing at least one very vigorous physical activity bout for the cardiorespiratory fitness deciles. The left panel shows the results for the metabolic equivalents (MET), and the right panel shows the oxygen uptake reserve (VO_2_R) value. The threshold for a moderate bout is 8.8 MET or 90% of VO_2_R. The *y*-axis shows the number of active hours, and the *x*-axis shows the maximal oxygen uptake (VO_2_max) of the decile. The blue line shows the men’s value, and the red line shows the women’s value. The vertical whiskers show a 95% confidence interval.

**Table 1 ijerph-20-06852-t001:** Number of participants (N) in each age group and mean (SD) maximal oxygen uptake (VO_2_max), body mass index (BMI), measured daily peak intensity in metabolic equivalents (MET) value and proportion of oxygen uptake reserve (VO_2_R) in cardiorespiratory fitness (CRF) tertiles broken down by sex and age group.

Sex and Age Group	N	CRF Group	VO_2_max (mL/kg/min)	BMI (kg/m^2^)	Peak MET	Peak VO_2_R
Men						
20–29	80	low	35.2 (5.1) ^MH^	28.0 (5.4) ^H^	5.9 (0.9) ^H^	55% (11%) ^H^
		med	42.4 (0.9) ^LH^	25.4 (3.2) ^H^	6.8 (1.5)	52% (13%)
		high	47.4 (2.2) ^LM^	22.9 (2.4) ^LM^	7.0 (1.8) ^L^	47% (13%) ^L^
30–39	205	low	35.1 (2.9) ^MH^	29.1 (4.3) ^MH^	5.5 (0.9) ^MH^	51% (11%)
		med	39.9 (1.1) ^LH^	25.7 (2.7) ^LH^	6.4 (1.6) ^L^	52% (15%)
		high	45.0 (2.3) ^LM^	24.5 (2.7) ^LM^	6.9 (1.9) ^L^	49% (16%)
40–49	288	low	31.4 (3.8) ^MH^	29.8 (4.8) ^MH^	5.5 (1.2) ^MH^	58% (18%) ^MH^
		med	38.1 (1.5) ^LH^	26.3 (2.9) ^LH^	6.0 (1.2) ^LH^	50% (12%) ^L^
		high	43.4 (2.3) ^LM^	24.4 (2.4) ^LM^	6.9 (1.8) ^LM^	51% (15%) ^L^
50–59	357	low	28.1 (3.8) ^MH^	29.9 (4.0) ^MH^	4.9 (0.8) ^MH^	57% (12%) ^MH^
		med	34.7 (1.4) ^LH^	26.5 (2.6) ^LH^	5.6 (1.2) ^LH^	51% (13%) ^L^
		high	40.0 (2.2) ^LM^	25.6 (2.3) ^LM^	5.9 (1.2) ^LM^	48% (11%) ^L^
60–69	517	low	25.1 (3.9) ^MH^	29.5 (4.1) ^MH^	4.7 (0.7) ^MH^	61% (15%) ^MH^
		med	31.3 (1.2) ^LH^	27.2 (3.0) ^LH^	5.0 (0.7) ^LH^	51% (9%) ^LH^
		high	36.8 (2.4) ^LM^	24.8 (3.1) ^LM^	5.6 (1.2) ^LM^	48% (12%) ^LM^
Women						
20–29	188	low	33.3 (4.4) ^MH^	26.9 (4.6) ^MH^	6.1 (1.1) ^H^	61% (14%) ^MH^
		med	39.3 (2.7) ^LH^	23.8 (2.9) ^LH^	6.6 (1.4) ^H^	55% (14%) ^L^
		high	44.0 (3.5) ^LM^	21.7 (2.4) ^LM^	6.9 (1.5) ^LM^	51% (12%) ^L^
30–39	326	low	31.4 (4.6) ^MH^	29.1 (4.4) ^MH^	5.6 (1.0) ^MH^	60% (16%) ^MH^
		med	37.5 (2.4) ^LH^	24.9 (2.6) ^LH^	6.3 (1.3) ^LH^	55% (14%) ^L^
		high	42.5 (3.0) ^LM^	22.8 (2.8) ^LM^	6.8 (1.7) ^LM^	53% (15%) ^L^
40–49	448	low	28.8 (4.3) ^MH^	30.7 (4.7) ^MH^	5.5 (1.0) ^MH^	64% (19%) ^MH^
		med	35.8 (2.3) ^LH^	25.2 (2.7) ^LH^	5.9 (1.1) ^LH^	53% (12%) ^L^
		high	40.9 (2.9) ^LM^	23.0 (2.3) ^LM^	6.8 (1.7) ^LM^	54% (15%) ^L^
50–59	515	low	26.1 (3.9) ^MH^	30.5 (3.9) ^MH^	4.9 (0.8) ^MH^	62% (16%) ^MH^
		med	32.7 (2.2) ^LH^	26.4 (2.7) ^LH^	5.5 (1.0) ^LH^	54% (12%) ^L^
		high	37.7 (2.9) ^LM^	24.3 (2.6) ^LM^	6.0 (1.2) ^LM^	51% (12%) ^L^
60–69	663	low	23.5 (3.8) ^MH^	30.3 (4.3) ^MH^	4.6 (0.6) ^MH^	65% (16%) ^MH^
		med	30.1 (1.6) ^LH^	26.2 (3.0) ^LH^	5.0 (0.7) ^LH^	53% (9%) ^LH^
		high	35.6 (2.5) ^LM^	23.7 (2.9) ^LM^	5.5 (1.0) ^LM^	49% (11%) ^LM^

The superscripted letter H denotes a significant (*p* < 0.05) difference for the sex and age group-specific high CRF group; the letter M stands for a significant difference for the medium CRF group, and the letter L stands for a significant difference for the low CRF group.

## Data Availability

Study data are not publicly available due to identifying patient data should not be shared. Upon reasonable request, de-identified data may be available from the corresponding author.

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
