# Peer review of "What Can One Minute of the Day Tell about Physical Activity?"

_ijerph, 2023, doi:10.3390/ijerph20196852_

Round 1
Reviewer 1 Report
The Vähä-Ypyä study and al. provided valuable information, and tries to answer the question if any physical activity could lead to cardiorespiratory fitness. Numerous studies showed that physical activity and structure exercise are beneficial for cardiorespiratory fitness and individuals with high VO2max are better protected against cardiovascular and metabolic diseases. However, this question has remained unanswered which type of physical activity or exercise would be beneficial? In addition, the authors find that people with VO2max around 30 ml/kg/min can be protect from strenuous physical activity during daily routines (individuals who are unable to avoid strenuous PA bouts).
In their discussion the authors explain and raise one important question:
“While the blue-collar workers are physically active for several hours each working day, they tend to have poorer cardiorespiratory fitness (CRF) and overall health compared to white-collar workers [28]. This raises the question of whether all types of PA are equally health-enhancing [27]. The recent WHO guidelines for physical activity and sedentary behavior acknowledge the need for further research on the impact of different domains of PA on health [5]. It is essential to consider that adults with high levels of occupational physical activity might benefit from more detailed recommendations of leisure-time physical activity”.
Vähä-Ypyä and al. studied a large number of men and women from 20 to 69 years old where they estimated their VO2max through 6 minutes’ walk and later on providing an accelerometer, where the subjects wore for a week. Then the daily peak MET value was recorded in both absolute values and relative to individual’s oxygen uptake reserve.
There is just one minor correction and two questions.
Correction: on Figure 4, Figure 5, Figure 6 where number is written nymber!
-Why there is such difference between men and women number?
-Line 104, in order to estimate VO2max why in men, it was predicted as authors wrote
“ the VO2max (in METs) was predicted from the walking distance in six minutes, age, body mass index (BMI), heart rate at the end of the test, and height. For women, the prediction was based on the walking distance in six minutes, body mass, and age. The accuracy of prediction is about 1 MET” and heart rate at the end of the test and height were not taken in consideration?
Author Response
We thank the reviewer for the beneficial feedback, which has contributed to the refinement of our manuscript.
Correction: on Figure 4, Figure 5, Figure 6 where number is written nymber!
The typos on the figures are now corrected.
-Why there is such difference between men and women number?
The men and women seem move at the same absolute intensity. However, in relative terms women have to utilize more of them aerobic capacity. It could be interesting to study who set the pace in the daily routines. This is now also pointed out in the discussion.
-Line 104, in order to estimate VO2max why in men, it was predicted as authors wrote
We have now added more details of the 6 min walk test into the manuscript. The heart rate was not a statistically significant predictor in the women’s equation and excluding it from men’s model would reduce the predictive power from 82% to 75% and increase the SEE from 3.6 to 3.9 ml/kg/min.
Reviewer 2 Report
Manuscript review; What can one minute of the day tell about physical activity?Journal: International Journal of Environmental Research and Public Health
General comments: This study assessed 1 wk accelerometer data from a large sample to test whether the relative and absolute level of daily PA varies based on participants’ CRF. Results show that those with higher CRF spend more time at high intensities; whereas, when expressed according to %VO2R, low fit adults spend more time in vigorous intensities due to their low CRF. Study is strengthened by a large diverse sample and comprehensive PA recording, although a limitation is the prediction of CRF rather than lab based determination.
Specific comments: Please consider the comments listed below concerning your submission; thank you.
Abstract. Please list mean age of all participants. Also, rewrite ln 17 to denote that less fit adults utilize a greater fraction of their aerobic capacity, as your data suggests that they are exercising at higher relative intensities.
Introduction; PA is a ‘blanket term’ here, are you referring to solely aerobic exercise, resistance training, lifestyle PA, etc.? I think it is important that you objectify EXACTLY the type of activity this term refers to, as otherwise, this is relatively vague. I say this as rarely does resistance training augment CRF, and it is clearly a popular mode of PA.
Ln 32 needs a reference citation for this definition of CRF.
Ln 61, what do you mean by ‘inverted?’ Tom Cruise in TopGun was flying ‘inverted’..do you mean opposite, etc. If so, please rephrase this text.
VO2R involves a measurement of resting HR as well as maximal HR; I do not believe the Methods text contains information pertaining to how these were identified.
Results are very clear and the text nicely accompanies and describes the various Figures, although there are various typos which I ask you to correct.
Ln 242, Discussion: What do you mean that those with CRF are ‘engaged in exercise bouts.’ This is a little confusing. And I think it is more clear and accurate to state that adults with low CRF utilize more (a greater fraction) of their aerobic capacity in daily activity…
Ln 271: Rather than stating that low CRF adults ‘cannot avoid strenuous activity,’ I think it is more accurate to state that daily PA requires a larger fraction of CRF in low active adults, ….The fact is, if these people avoided voluntary movement, they could avoid strenuous activity.
You do not really state what the implications are of low fitness adults requiring more vigorous efforts (i.e. higher %VO2R) to engage in lifestyle PA; are they prone to fatigue? Basically, what are the effects or ‘take home message’ of this result?
Ln 302-303; Please revise this and specifically identify the ‘action’ you refer to here.
Also, this study would benefit from a limitation section.
It is ok and certain areas need attention so the text is more clear and more proper/appropriate language is used.
Author Response
We thank the reviewer for the valuable comments and insightful suggestions. In response, we have modified the manuscript according to the comments. Please, find these revisions and accompanying comments detailed below.
Abstract. Please list mean age of all participants. Also, rewrite ln 17 to denote that less fit adults utilize a greater fraction of their aerobic capacity, as your data suggests that they are exercising at higher relative intensities.
We have added the age and rewrite the line 17.
l
Introduction; PA is a ‘blanket term’ here, are you referring to solely aerobic exercise, resistance training, lifestyle PA, etc.? I think it is important that you objectify EXACTLY the type of activity this term refers to, as otherwise, this is relatively vague. I say this as rarely does resistance training augment CRF, and it is clearly a popular mode of PA.
PA is a broad term and can lead to confusing assumptions. In this study the PA means just aerobic part of the PA. This is now stated more clearly.
Ln 32 needs a reference citation for this definition of CRF.
The definition of CRF has now the citation.
Ln 61, what do you mean by ‘inverted?’ Tom Cruise in TopGun was flying ‘inverted’..do you mean opposite, etc. If so, please rephrase this text
The term ‘inverted’ is now changed to term ‘complex’. Anyhow, the results could be interpreting as flying upside down. In relative terms, the participants with high CRF accumulated the least amount of moderate to vigorous PA time.
VO2R involves a measurement of resting HR as well as maximal HR; I do not believe the Methods text contains information pertaining to how these were identified.
The definition and calculation of the VO2R is now presented more clearly. It is based on the predicted VO2max value from 6 min walking test and estimated peak MET value from the accelerometry. Therefore, the HR is not needed.
Ln 242, Discussion: What do you mean that those with CRF are ‘engaged in exercise bouts.’ This is a little confusing. And I think it is more clear and accurate to state that adults with low CRF utilize more (a greater fraction) of their aerobic capacity in daily activity
The line ‘engaged in exercise bouts’ is now rephrased and stated more clearly.
Ln 271: Rather than stating that low CRF adults ‘cannot avoid strenuous activity,’ I think it is more accurate to state that daily PA requires a larger fraction of CRF in low active adults, ….The fact is, if these people avoided voluntary movement, they could avoid strenuous activity.
The line ‘cannot avoid strenuous activity’ is now expressed as ‘they have to utilize high fraction of their aerobic capacity throughout the day’.
You do not really state what the implications are of low fitness adults requiring more vigorous efforts (i.e. higher %VO2R) to engage in lifestyle PA; are they prone to fatigue? Basically, what are the effects or ‘take home message’ of this result?
The implication the low fitness and effects of it are now stated with following sentences: “Based on our current study, a VO2max of approximately 30 ml/kg/min appears to protect from strenuous PA during daily routines. Low fitness individuals have to work at a higher percentage of their aerobic capacity to perform the same activities as their more fit counterparts, which can put additional strain on their cardiovascular system [33]. Also, they might find if challenging to sustain vigorous efforts during physical activity and become fatigued more quickly [34]. Nevertheless, it is advisable to take proactive actions to mitigate the decline in CRF well before reaching this critical threshold.”
Ln 302-303; Please revise this and specifically identify the ‘action’ you refer to here.
Related to the detailed recommendations, we have now added following sentences: “While they may already have high levels of mandatory physical activity, encouraging them to increase their activity levels further could potentially result in chronic strain, injuries, or fatigue. [30]. They could benefit from the optimal amount of PA, sedentary behavior and recovery to improve their CRF level [29]. Occupation-specific recommendations for CRF may also be necessary. “
Also, this study would benefit from a limitation section.
We have added following limitations: “Furthermore, it's important to acknowledge that the precision of our VO2max and peak MET estimations could be considered a limitation of this study. Achieving greater precision in VO2max measurements would require a test done until exhaustion and involving breath-by-breath measurement of the gas concentrations of inspired and expired air [22]. Enhancing the precision of the peak MET estimation could involve individual calibration [27]. Nevertheless, considering the large-scale sample size in our present study, the methods employed can be viewed as both feasible and cost-effective.”
It is ok and certain areas need attention so the text is more clear and more proper/appropriate language is used.
We have corrected typos and rephrased some words.
Round 2
Reviewer 2 Report
I am pleased with the changes made to the paper as well as the Authors' responses to my initial concerns.